# Type 2 Diabetes Mellitus, Non-Alcoholic Fatty Liver Disease, and Metabolic Repercussions: The Vicious Cycle and Its Interplay with Inflammation

**DOI:** 10.3390/ijms24119677

**Published:** 2023-06-02

**Authors:** Rafał Frankowski, Mateusz Kobierecki, Andrzej Wittczak, Monika Różycka-Kosmalska, Tadeusz Pietras, Kasper Sipowicz, Marcin Kosmalski

**Affiliations:** 1Students’ Research Club, Department of Clinical Pharmacology, Medical University of Lodz, 90-153 Lodz, Poland; rafal.frankowski@stud.umed.lodz.pl (R.F.); mateusz.kobierecki@stud.umed.lodz.pl (M.K.); andrzej.wittczak@stud.umed.lodz.pl (A.W.); 2Department of Clinical Electrocardiology, Medical University of Lodz, 92-213 Lodz, Poland; monika.rozycka-kosmalska@umed.lodz.pl; 3Department of Clinical Pharmacology, Medical University of Lodz, 90-153 Lodz, Poland; tadeusz.pietras@umed.lodz.pl; 4Department of Interdisciplinary Disability Studies, The Maria Grzegorzewska University in Warsaw, 02-353 Warsaw, Poland; ksipowicz@aps.edu.pl

**Keywords:** non-alcoholic fatty liver disease, type 2 diabetes mellitus, chronic inflammation, oxidative stress, cytokines, pathogenesis

## Abstract

The prevalence of metabolic-related disorders, such as non-alcoholic fatty liver disease (NAFLD) and type 2 diabetes mellitus (DM2), has been increasing. Therefore, developing improved methods for the prevention, treatment, and detection of these two conditions is also necessary. In this study, our primary focus was on examining the role of chronic inflammation as a potential link in the pathogenesis of these diseases and their interconnections. A comprehensive search of the PubMed database using keywords such as “non-alcoholic fatty liver disease”, “type 2 diabetes mellitus”, “chronic inflammation”, “pathogenesis”, and “progression” yielded 177 relevant papers for our analysis. The findings of our study revealed intricate relationships between the pathogenesis of NAFLD and DM2, emphasizing the crucial role of inflammatory processes. These connections involve various molecular functions, including altered signaling pathways, patterns of gene methylation, the expression of related peptides, and up- and downregulation of several genes. Our study is a foundational platform for future research into the intricate relationship between NAFLD and DM2, allowing for a better understanding of the underlying mechanisms and the potential for introducing new treatment standards.

## 1. Introduction

NAFLD refers to a condition characterized by hepatic steatosis, which involves the accumulation of fat in at least 5% of hepatocytes in the liver. This condition occurs without any identifiable secondary causes such as alcohol use or hepatitis C virus infection [1]. NAFLD has emerged as the leading liver disease in terms of prevalence, accounting for approximately 32.4% of cases and exhibiting a rising trend in recent years. In terms of gender distribution, NAFLD is more frequently diagnosed in men (39.7%) than women (25.6%) [2].

As NAFLD progresses, it may advance to a more severe condition known as non-alcoholic steatohepatitis (NASH). In NASH, additional components such as liver damage, inflammation, and fibrosis become evident, indicating a more pronounced level of liver injury. The diagnosis of NASH can be made based on several critical indicators, including the initial statement of at least 5% liver steatosis (fat accumulation in liver cells). Additionally, varying degrees of hepatic fibrosis (abnormal accumulation of fibrous tissue), hepatocyte ballooning degeneration (swelling and damage to liver cells), and lobular inflammation (inflammation in the lobules of the liver) are considered significant factors in confirming a diagnosis of NASH [3,4,5,6,7]. In 2015, NASH accounted for about 20% of NAFLD cases in the United States, and that percentage is expected to increase shortly [8]. NASH poses a significant health concern, impacting approximately 1.5–6.5% of the general population. It ranks among the primary causes of liver cirrhosis in the United States, highlighting the potential for severe liver damage as the disease progresses. The early detection and appropriate management of NASH are crucial to mitigate these risks [9,10].

NAFLD, a metabolic disorder, connects with other metabolic defects such as obesity, dyslipidemia, insulin resistance, and diabetes mellitus (DM). Nevertheless, emphasizing that this phenomenon occurs even in individuals with average weight is crucial. Among the reasons for the development of NAFLD in lean people high alcohol intake, lipodystrophy, polycystic ovary syndrome, deficiency of growth hormone, or diets rich in fructose and fat can be specified. Nevertheless, NAFLD can be caused by drugs such as amiodarone, tamoxifen, or methotrexate and even may be related to genetic disorders, for example, lysosomal acid lipase deficiency [3,11,12]. The growing prevalence of the diseases above points to an anticipated rise in NAFLD cases. NAFLD, in turn, serves as a trigger, heightening the risk of developing other disorders. Notably, NASH, cirrhosis, and hepatocellular carcinoma take the forefront. NASH specifically predisposes individuals to cirrhosis in about 20% of cases and hepatocellular carcinoma, cardiovascular diseases, and other malignancies [3,4,5,6,7]. 

In the literature, there is an extensive exploration of the connections between NAFLD and the risk of several cancers, such as colorectal, kidney, uterine, and bladder cancer. However, the highest risk is observed for hepatocellular carcinoma. The possible mechanism of the elevated risk of cancer development in NAFLD subjects may be related to chronic inflammation and visceral adiposity, but this mechanism is still under research [13]. Additionally, researchers have documented an elevated risk of developing DM2 [14], colonic diverticulosis [15], cardiovascular diseases [16,17], and chronic kidney disease [18,19] among individuals with NAFLD. NAFLD is acknowledged to increase mortality in affected patients, primarily due to the underlying mechanisms rooted in atherosclerosis and cardiovascular diseases. These mechanisms are related to metabolic syndrome, which is connected with hypertension or proatherogenic lipid profile [20,21].

A recent proposal suggests using metabolic-dysfunction-associated fatty liver disease (MAFLD) instead of NAFLD, as it provides a broader perspective and emphasizes the connection between this condition and metabolic dysfunction. Clinicians can establish the diagnosis of MAFLD when liver steatosis coexists with at least one of the following criteria: overweight, diagnosis of DM2, or the presence of at least two out of five metabolic syndrome risk factors [22,23]. 

DM comprises a group of metabolic diseases that exhibit chronic hyperglycemia. These conditions stem from defects in either insulin secretion, insulin action, or both [24]. 

The worldwide prevalence of DM in 2021 (in adults aged 20–79 years) was estimated at 10.5% (536.6 million people) regardless of its type. It is crucial to note that the prevalence of all types of DM is on the rise. According to projections, by 2045, approximately 783.2 million individuals, which accounts for 12.2% of adults aged 20–79 years, are expected to be living with DM, including all types of this disease [25]. DM2 accounts for 90–95% of all DM types. The development of DM2 arises from a gradual decline in β-cell insulin secretion, primarily occurring in the presence of insulin resistance and metabolic syndrome and without the involvement of autoimmune factors [26]. Although various genetic and environmental factors can result in the progressive loss of β-cell mass and/or function, once hyperglycemia occurs, people with all forms of diabetes are at higher risk for developing the same chronic complications. The inflammatory process is involved in disturbing the β-cell proper function and is one of the factors leading to microvascular (e.g., diabetic nephropathy, retinopathy, and neuropathy) and macrovascular (e.g., coronary artery disease) DM complications. High levels of proinflammatory cytokines may lead to hyperglycemia [27,28].

Inflammation is a complex physiological response to the antigens such as foreign organisms, particles from another individual, cancer cells, or even cells of the host. The response to a foreign organism’s antigens is crucial for the body to stay healthy. Nevertheless, disorders of the immune system and its over-reactivity to self-antigens, which the body normally distinguishes from foreign ones, lead to an excessive inflammatory reaction and autoimmune diseases [29]. During inflammatory response, different changes can be observed such as tissue damage and its decreased function, the presence of increased numbers of immune cells, and the local production of cytokines and chemokines [30].

Additionally, there is a significant correlation between adipose tissue and low-grade inflammation, particularly in individuals who are obese. This relationship highlights the link between excess body fat and chronic, low-level inflammation [31]. 

Studies have indeed demonstrated that in individuals with morbid obesity, proinflammatory genes are more expressed in subcutaneous adipose tissue than visceral adipose tissue. This observation suggests that the subcutaneous fat layer, which lies just beneath the skin, may significantly influence inflammation in cases of severe obesity [32]. 

Indeed, research has provided evidence of a strong connection between visceral adipose tissue and inflammation [33]. A notable study demonstrated decreased C-reactive protein (CRP) levels after omentectomy (surgical removal of the omentum) in individuals who underwent the Roux-en-Y gastric bypass procedure [34]. However, data in this field are inconsistent, as other studies did not reveal a decrease in inflammatory markers after omentectomy [35]. This possible reduction in CRP levels following the removal of visceral adipose tissue underscores this type of fat’s significant role in promoting inflammation. Inflammation is traditionally divided into acute and chronic processes. The latter is a cause of the long-term overactivity of cytokines and immune cells. Recently, it has been postulated that chronic inflammation-related diseases are the main cause of death around the world. It is a well-known factor that plays a key role in the development of complex diseases such as DM2 and NAFLD [28,36]. Insulin resistance (IR) is another factor that could be described as a connector between the pathogenetic path of NAFLD and DM2, as it plays a role in the development of both [37,38]. 

The relationship between NAFLD and DM2 is complex and can be defined as bidirectional. DM2 development can be promoted via NAFLD, and on the other hand, DM2 is recognized as an NAFLD-promoting factor [3,39]. Insulin resistance, a decrease in antioxidant capacity, and an increase in free radical oxidation products are considered factors leading to the development of NAFLD [40]. During inflammation, Kupffer cell activation increases the secretion of inflammatory cytokines such as interleukin (IL)-6, tumor necrosis factor α (TNF-α), and IL-1β, which in turn enhance the body’s immune response to eliminate damaged cells and induce oxidative stress [41]. Oxidative stress could be described as a disorder of stability between antioxidant and pro-oxidant processes [42].

The interplay between NAFLD, DM2, and chronic inflammation is key to understanding the complicated pathogenesis of these conditions. What follows this is the introduction of new diagnostic and therapeutic standards, which can improve patients’ conditions and decrease costs related to complications of these diseases.

## 2. Materials and Methods

### 2.1. Focal Question

The demonstration of the connections between the chronic inflammatory process and pathogenesis of NAFLD and DM2 was set as a major aim of this review.

### 2.2. Language

Studies published in English were included in this analysis.

### 2.3. Databases

The PubMed database according to Preferred Reporting Items for Systematic Reviews and Meta-Analyses (PRISMA) guidelines [43] was searched (Appendix A), the PRISMA flow diagram indicating the number of articles included and excluded was prepared (Appendix A).

### 2.4. Study Extraction

In relation to publication date, the evaluated papers were limited to those published since 2000; more than 50% of searched papers have been published since 2019. Mainly, we searched clinical trials and randomized controlled trials. The final search was conducted on 25 March 2023. In order to find matching articles, we used a combination of Medical Subject Heading (MeSH) terms and specific search terms in variations, including “non-alcoholic fatty liver disease”, “type 2 diabetes mellitus”, “chronic inflammation”, “pathogenesis”, and “progression”. The relevant works fulfilling the following inclusion criteria were selected: studies in English that focus on chronic inflammation, non-alcoholic fatty liver disease, non-alcoholic steatohepatitis, and type 2 diabetes mellitus, and their pathogenesis and relationships. All the selected studies were then downloaded and analyzed.

### 2.5. Data Extraction

Initially, titles, abstracts, or full texts were reviewed; subsequently, full copies were analyzed. In relation to the prepared and tested template, data on NAFLD and DM2 pathogenesis in the context of the inflammatory process and the interrelationships between them were extracted from the articles. Analyses were performed using Microsoft Office 365, CiteSpace (v.6.1. R2), and VOSviewer (v.1.6.18) to conduct a bibliometric and visual analysis of all data.

### 2.6. Quality Assessment

The final assessment and verification were carried out by experienced researchers (M.K., M.R.-K., K.S., and T.P.).

## 3. NAFLD and DM2 Coexistence

As the pathogenetic factors leading to the expansion of both NAFLD and DM2 are similar, including dyslipidemia, obesity, genetics, environment, lifestyle factors, and insulin resistance, it is not surprising that they often coexist [44]. As NAFLD and DM2 share similar physiopathological pathways, one may precede and/or promote the other. People with NAFLD are at greater risk of developing DM2; at the same time, DM2 leads to the onset or progression of NAFLD [45]. Hepatic IR associated with NAFLD, which may be secondary to the inflammation caused by the deposition of toxic metabolites from triglycerides (TGs) in tissues such as the pancreas or liver, is responsible for the development of DM2 in NAFLD populations [46]. Among the many functions of insulin regulation of TGs, fatty acid metabolism, hepatic gluconeogenesis, and glycogenolysis can be highlighted. Hyperinsulinemia related to DM2, which occurs to overcome IR, results in alterations in the hepatic clearance of insulin and promotes NAFLD development [47]. The prevalence of DM2 in patients with NAFLD depends on the severity of NAFLD, ranging from 9.8% in mild NAFLD to 17.8% in moderate-to-severe NAFLD [46]. The estimated prevalence of NAFLD in patients with DM2, however, is as high as 75% [46]. The analyzed data suggest that although NAFLD and DM2 may occur independently, there is an intricate relationship between these diseases. Understanding the pathomechanism of the codevelopment of NAFLD and DM2 is still evolving [46,47,48].

The pathogenesis of DM2 and NAFLD is intricate and not fully understood; however, in the development of both, many common factors can be found. Similar causes of the pathogenesis of these diseases include, e.g., genetic predispositions [49,50], environmental and lifestyle influence [50], intestinal microbiome alterations [51,52], IR [53], or disturbances in lipid and glucose metabolism [54]. The development of NAFLD is strongly associated with hepatic IR [44]. Hepatic IR is a common feature that predisposes individuals to compensatory hyperinsulinemia and is also followed by consequent pancreatic-cell dysfunction and the development of DM2 [55]. Nevertheless, the association of NAFLD with systemic IR is a matter of debate. Indeed, the strong association between NAFLD and systemic IR suggests that NAFLD may be both a marker and a promoter of systemic IR [56]. NAFLD is responsible for the exacerbation of glucose intolerance and IR in multiple ways, including alterations in fatty acids’ β-oxidation and elevating levels of proinflammatory cytokines [48,57]. On the other hand, DM2 is one of the risk factors for liver fibrosis in NAFLD and its progression to NASH [52,58].

## 4. Inflammatory Mechanisms Underlying NAFLD and DM2 Pathogenesis

Metabolic disorders such as NAFLD have complex pathogenesis. In fact, metabolic syndrome, insulin resistance, obesity, alterations in lipid and glucose metabolism, high caloric intake, increased hepatic de novo lipogenesis, alterations in the gut microbiota, inflammatory processes, and oxidative stress are all associated with NAFLD [21,59]. Genetic factors also are considered risk factors for NAFLD occurrence and progression [58]. For example, the non-synonymous rs738409 C/G variant in the PNPLA3 gene (patatin-like phospholipase domain-containing 3) is regarded as the major genetic component of NAFLD and NASH. Other genes reported to be associated with NAFLD include PPP1R3B (protein phosphatase 1, regulatory subunit 3B), FDFT1 (farnesyl-diphosphate farnesyltransferase), ERLIN1 (ER lipid raft-associated 1), LTBP3 (latent-transforming growth factor beta) and PARVB (parvin beta) [60]. Genetic factors may also influence the effects of treatment. For example, according to Vilar-Gomez et al., high intakes of n-3 PUFAs (polyunsaturated fatty acids), total isoflavones, methionine, and total choline are associated with a reduced risk of fibrosis severity in carriers of the rs738409 variant in the PNPLA3 gene, thus pointing to the prospect of dietary treatment [61].

Many risk factors are known to influence the complex pathogenesis of DM2 [62]. These include modifiable factors such as diet, tobacco use, and physical activity [62]. There are also non-modifiable elements such as age, ethnicity, and genetics [62]. A large number of genes with reported altered deoxyribonucleic acid (DNA) methylation patterns have been linked with the pathogenesis of DM2 [63]. For example, the expression of the insulin (INS) gene is regulated through DNA methylation [64]. In the study by Yang et al. [64], four CpG sites within the insulin gene promoter showed increased DNA methylation in pancreatic islets from patients with DM2, compared with non-diabetic donors [64]. Inflammation is thought to be the primary trigger of DM2 [65]. The inflammatory process is induced by the aforementioned risk factors and through the chronic activation of proinflammatory cytokine pathways in the tissues targeted by insulin action, such as adipose tissue, muscle mass, and the liver [65]. For example, in terms of diet, red meat consumption has been shown to have proinflammatory properties [62]. In addition, high serum glucose levels, oxidized low-density lipoproteins (LDLs), free fatty acids (FFAs), and cholesterol act as endogenous damage-associated molecular patterns in diabetes and cause inflammasome activation [66].

Oxidative stress, endothelial dysfunction, and inflammation are proposed to be involved in NAFLD pathogenesis [67]. In addition, inflammation and oxidative stress have been postulated to exacerbate NAFLD and its progression to NASH [41]. For this reason, these factors are the subject of research aimed at finding a possible treatment for NAFLD [68]. As mentioned above, similar inflammatory factors may be associated with the pathogenesis of DM2 [27]. 

### 4.1. Inflammation as a Pathway of NAFLD Progression to NASH

The role of inflammation in exacerbating NAFLD state was previously mentioned. In the liver tissue, we can highlight macrophages named Kupfer cells, which are involved in the inflammatory process. These cells are responsible for the release of chemokines and cytokines in response to liver damage. Kupffer cells are the largest population of resident tissue macrophages in the liver [69]. The functional plasticity of macrophages is driven by their immunological environment, which can shape their properties through a wide spectrum of phenotypes, with classical (M1) or alternative (M2) representing the extremes [70]. Inflammation driven by M1 macrophages is counterbalanced by alternatively polarized M2 macrophages that promote the resolution of inflammation and tissue repair [70]. Traditional inducers of macrophage polarization to M2 include IL-4, IL-13, and transforming growth factor beta (TGF-β) [71]. M1 macrophages are known as proinflammatory macrophages because they can secrete many proinflammatory cytokines, such as IL-1β, inducible nitric oxide synthase (iNOS), and TNF-α [72]. It was also revealed that the polarization of M1 and M2 cells depends on 5′-adenosine monophosphate (AMP)-activated protein kinase (AMPK) activation via calcium-/calmodulin-dependent protein kinase kinase-β (CaMKKβ) [73]. Another study reported that M1 and M2 cell levels are affected by the activation of peroxisome proliferator-activated receptor gamma (PPAR-γ) [74]. Research showed that M1 macrophage polarization markedly increases during the development of MAFLD and NASH [72].

Under normal conditions, there is a balance between Kupffer cell types, and their net effect is beneficial. However, when this balance is disturbed, due to the imbalanced secretion of IL-1β, IL-6, IL-12, IL-18, TNF-α, and chemokine (C–C motif) ligand 2 (CCL2) chemokine, liver function is impaired, and NAFLD develops [59,75]. In NASH, the inflammatory response to cellular necrosis induces the progressive release of PDGF (platelet-derived growth factor), TGF-β, TNF-α, and other inflammatory factors, such as IL-1, by resident immune cells [76]. These inflammatory signals result in the activation and proliferation of hepatic stellate cells (HSCs) and induce the differentiation of HSCs into myofibroblasts, further driving extracellular matrix (ECM) synthesis and ultimately liver fibrosis [76,77]. It has been found that the induction of HSCs is associated with K_Ca_3.1 and K_Ca_2.3, which are components of a heterotetrameric voltage-independent potassium channel. Furthermore, it has been observed that oxidative stress upregulates these proteins, suggesting their involvement in the development of liver disease [78].

Duan et al. [79] conducted meta-analyses to determine the association of 16 inflammatory cytokines with NAFLD. Significant associations were found between NAFLD and levels of CRP, IL-1b, IL-6, TNF-α, and intercellular adhesion molecule-1 (ICAM-1). In contrast, no significant associations between interferon-gamma (IFN-γ), insulin-like growth factor II (IGF-II), IL-2, IL-4, IL-5, IL-7, IL-8, IL-10, IL-12, monocyte chemoattractant protein 1 (MCP-1), and TGF-β were found [79]. On the contrary, TGF-β1 plays a significant role in liver fibrosis by activating HSCs and promoting the generation of the extracellular matrix (ECM). However, it is essential to note that the activation of TGF-β1 can be counteracted or reduced by the presence of bone morphogenetic protein-7 (BMP-7). This suggests that BMP-7 may have a potential inhibitory effect on TGF-β1-mediated liver fibrosis [80]. The investigation carried out by Duan et al. [79] did not establish definitive evidence regarding the connections between IFN-γ and NAFLD. However, the study did reveal that the administration of IFN-γ in a rodent model resulted in a reduction in liver fibrosis. This reduction exhibited a dose-dependent relationship and could potentially be linked to the suppression of ECM synthesis [81].

It was also observed that in comparison to healthy subjects, patients diagnosed with NAFLD had elevated levels of soluble vascular cell adhesion molecule-1 (sVCAM-1), which is closely associated with endothelial dysfunction. SVCAM-1 is correlated with high-sensitivity C-reactive protein (hs-CRP) levels, which means that inflammation is associated with endothelial dysfunction [67]. In another study, it was revealed that NAFLD is related to the serum level of fibroblast growth factor 21 (FGF21) [82]. FGF21 is a signaling molecule that has a role in regulating lipid metabolism and cellular function, especially in the absence of nutrients [83]. In mice, the downturn of this cytokine caused fat accumulation in the liver and its friability [84]. 

Oxidative stress emerges as a pivotal factor of significance in liver function. Alkhouri et al. [42] proposed the use of oxidative stress exponents as a distinguishing parameter between NAFLD and NASH in an OxNASH score test [42]. It is believed that oxidative stress is a crucial factor in the development of NAFLD and the progression of the disease to NASH [85]. The progression to NASH through oxidative stress is thought to occur by stimulating Kupffer immune cells residing in the hepatic sinusoid. This stimulation leads to the activation of redox-sensitive transcription factors such as NF-κB and AP-1. Consequently, these transcription factors induce the expression and release of proinflammatory cytokines, chemokines, adhesion molecules, and fibrogenic mediators [86].

Accumulating evidence suggests the involvement of hepatic B lymphocytes in the development of NAFLD and its progression to NASH [87]. Individuals with NAFLD or NASH have more intrahepatic B cells and more circulating IgG that recognizes oxidative-stress-derived epitopes [87]. B cells have been associated with inflammation and fibrosis in mouse models [88]. The cause of B-cell activation in patients with NAFLD remains unknown; however, there is a hypothesis that oxidative stress and alterations in the gut microbiota cause B-cell differentiation [89]. 

Extensive molecular investigations have provided compelling evidence establishing a strong association between gene expressions and the progression of NAFLD to NASH, ultimately leading to hepatic cirrhosis. Notably, specific genes such as IL-1β and TGF-β1, matrix metalloproteinase 9 (MMP9), and matrix metalloproteinase 14 (MMP14), as well as ligands of chemokines, including CCL2 and chemokine (C–X–C motif) ligand 1 (CXCL1), have been meticulously identified and associated with these pathological processes. Crucially, the regulation of these genes is governed by the NOD-like receptor protein 3 (NLRP3) inflammasome (NLRP3i). The activation of these genes triggers a cascade of events involving inflammatory processes, lipid metabolism regulation, and remodeling of the extracellular matrix [90]. NLRP3i is upregulated by cathepsin B, for which the role of NAFLD pathogenesis has been established. A rodent model revealed that the inhibition of cathepsin B resulted in a decrease in NLRP3i activity in Kupffer cells [90]. Additionally, a decline was observed in liver inflammation and IL-1β and IL-18 levels. Improvement in hepatocyte ballooning and decreased caspase-1 activity are also related to the inhibition of cathepsin B [91]. It has also been revealed that NLRP3i is activated by heat-shock protein 90 (HSP90) [92].

In recent years, single-nucleotide polymorphisms (SNPs) are being studied for their role in an increased risk of NAFLD development. These SNPs include glucokinase regulatory protein (*GCKR*) rs780094, patatin-like phospholipase domain-containing 3 *(PNPLA3)* rs738409, and peroxisome proliferator-activated receptor-γ coactivator (PGC)-1α gene *(PPARGC1A)* rs8192678 genes. Some genetic variants of the listed SNPs have been shown to confer susceptibility to NAFLD [93,94]. When it comes to TGF-β1 levels concerning healthy individuals, a study by Hasegawa et al. [95] revealed increased plasma concentration in subjects with NASH. Interestingly, no differences in TGF-β1 were found between NAFLD-diagnosed patients and healthy subjects. TGF-β is known for its profibrotic effect on the liver. The authors suggested that it could be used as a marker for differentiating between NASH and NAFLD. Researchers have revealed an association between C–C chemokine receptor type 2 (CCR2) and type 5 (CCR5) and their ligands (CCL2 and CCL5) concerning liver fibrosis [96,97]. These receptors and ligands are instrumental in triggering the inflammatory response and facilitating the migration of immune cells, ultimately leading to the development of hepatic fibrosis. However, significant advancements in treating liver fibrosis have been achieved by administering an oral antagonist targeting CCR2 and CCR5 known as cenicriviroc [10]. This treatment has demonstrated notable improvement in liver fibrosis and the regression of systemic inflammatory markers, including IL-6, hs-CRP, IL-1β, and fibrinogen [10]. 

Interestingly, it was shown that TNF-α-308 G/A and IL-10-1082 G/A genotypes are associated with appropriate levels of cytokines in plasma. They were found to be statistically significantly more frequent than in healthy controls. These genotypes were correlated with a decrease in IL-10 and an increase in TNF-α levels in plasma. As an anti-inflammatory cytokine and its altered genetic variance, IL-10 may be a cause of varied inflammatory responses and NAFLD progression to NASH [98].

Tao et al. [3] revealed the beneficial effects of a 12-week inhalation treatment in patients with NAFLD using a mixture of hydrogen and oxygen. Improvement in liver histopathology, a reduction in the amount of liver fat, and liver inflammation were revealed due to this procedure. Moreover, mixture inhalation resulted in a downturn of systemic inflammatory parameters such as TNF-α and IL-6. Improvement in oxidative stress was also observed [3].

Recently, researchers have reported on the effect of microRNAs (mi-RNA) in NAFLD, especially mi-RNA155, which is involved in the inflammatory response, and its upregulation was observed in the liver of NAFLD subjects [99].

The importance of gut microbiota in human health is well established, and alterations in the microbiome have been linked to the pathogenesis of NAFLD and DM2. As indicators of this interplay, we can point to the breakage of the intestinal barrier, the translocation of microbiota, and inflammation [38]. Studies have been conducted in this context. One of these studies has proven beneficial outcomes in correcting the microbial state of rodents guts, which results in a decreased expression of inflammatory parameters such as TNF-α, monocyte chemoattractant protein-1 (Mcp-1), collagen type I alpha 1 (Col1α1), collagen type 1 alpha 2 (Col1α2), and TGF-β1 [7].

It is known that obesity is a factor that induces systemic inflammation and reduces anti-inflammatory pathways. One of these pathways is the heat-shock transcription factor-1 (HSF1)–70 kDa family of heat-shock protein (HSP70) (HSF1-HSP70) axis, and a reduction in this pathway was observed in the liver and adipose tissue of NAFLD-diagnosed patients. Consequently, obesity exacerbates the development of NAFLD [100]. It has been shown that weight loss reduces the serum levels of TNF-α, IL-6, and IL-8 and improves the subject’s health state, which could be measured by enhancement in alanine aminotransferase (ALT), AST, IR factors, and lipid maintenance [101]. A previous study revealed that regular natural killer (NK) cells levels in subjects with NAFLD are lower than in healthy subjects, but the level of impaired phenotype NK cells Siglec-7−CD57+PD-1+CD56^dim^ was higher in NAFLD patients’ sera. The authors suggest that there may be some association between NK-cell function and NAFLD progression [102]. The pathogenesis of NAFLD and NASH in terms of inflammation is presented in Figure 1.

### 4.2. Inflammation as a Factor in Insulin Resistance 

As the importance of IR was discussed in previous sections, the molecular mechanisms of IR interplay with inflammation will be the theme of this section. In a previous study, Cifarelli et al. [103] confirmed the relationship between low levels of oxygenation in adipose tissue and its production of inflammatory cytokines, leading to a decrease in insulin sensitivity. It is important to note that obesity does not always indicate a poor metabolic state of the system. It may be associated with genetic and hereditary factors responsible for the angiogenesis of excessive adipose tissue [103]. The mentioned study is in line with the evidence indicating that a decrease in oxidative stress results in an improvement in IR [104]. 

According to the analyzed data, it can be concluded that poor vascularity in the system may lead to similar changes. The hypothesis about poorer tissue oxygenation and its connection with IR is confirmed by the fact that intermittent hypoxia adversely affects insulin sensitivity. Hypoxia may play a role in obstructive hypoventilation syndrome (OHS). A study performed among obese subjects with OHS revealed that continuous positive airway pressure (CPAP) therapy improved metabolic parameters, insulin sensitivity, and the Homeostatic Model Assessment for Insulin Resistance (HOMA-IR) scores [105]. 

It has been proven that improvement in insulin resistance results in the regression of blood pressure, vascular inflammation, and enhancement of glycemia and lipid management. Moreover, these changes were associated with a decrease in expression in TNF-α and thioredoxin-interacting protein (TXNIP), which are known as mediators of IR. Additionally, decreases in MCP-1 and IL-8 levels are postulated as effects of improved insulin sensitivity [104,106]. Researchers have different views regarding the action of IL-6 on IR, some suggest it has beneficial effects, while others point to its negative impact on IR [107]. On the one hand, there are indeed studies suggesting the beneficial effects of IL-6 on IR. Benrick et al. [108] performed an experiment in mice and showed that endogenous IL-6 contributes to the exercise-induced increase in insulin sensitivity [108]. Using a mouse model, Mauer et al. [109] demonstrated that IL-6 plays a homeostatic role in limiting obesity-associated insulin resistance and inflammation by reducing proinflammatory gene expression and increasing IL-4 responsiveness in macrophages [109]. On the other hand, Rehman et al. [110] explored the relationship between IL-6 stimulation and insulin resistance. According to these authors, IL-6 causes IR by impairing the phosphorylation of the insulin receptor and insulin receptor substrate 1 by inducing the expression of SOCS-3 (suppressor of cytokine signaling 3), a potential inhibitor of insulin signaling [110]. All in all, the question of the effect of IL-6 on IR seems to be still unclear and therefore needs further research.

### 4.3. Inflammation in the Pathogenesis of Type 2 Diabetes Mellitus 

In the pathogenesis of DM2, pancreatic β-cell dysfunction can be distinguished during its course through the detection of a gradual decrease in the number of β cells [111,112,113]. DM2 develops when pancreatic cells fail to produce enough insulin to overcome insulin resistance [30]. Islet cell damage is caused by inflammation and oxidative stress. They are also the main initiators of altered insulin-signaling pathways [104]. In the pancreas of patients with type 2 diabetes mellitus, β cells producing IL-1β were observed, which is associated with increased glucose levels [114]. This inflammatory cytokine is involved as an effector molecule in the destruction of β cells. The pancreas is susceptible to the action of this cytokine due to the presence of multiple IL-1 receptors [115,116]. It is known that IL-1β secretion, macrophage infiltration, and caspase-1-related pyroptosis are related to NLRP3i activity [117,118,119]. In a mouse model, it was found that NLRP3i in pancreatic β cells could be activated by the environmental endocrine disruptor PCB118 via the oxidative stress pathway, leading to pancreatic β-cell inflammation. Increased levels of CCL-2, IL-1β, IL-6, IL-18, and caspase-1 were also observed in pancreatic β cells after PCB118 exposure [120]. It was observed that among diabetic subjects, a decrease in this inflammasome activity leads to an improvement in glycemic-related parameters and even prevents DM2 development by having protective effects on pancreatic β cells [121,122].

In terms of DM2 pathogenesis, the role of IL-6 remains unclear. Interesting conclusions were drawn in a study highlighting the influence of IL-6 on the mechanism of glucagon-like peptide 1 (GLP-1) secretion and action. GLP-1 takes part in insulin excretion through its potentializing effects on β cells. While under stable conditions, it equally increases and lowers blood sugar levels, in subjects with an imbalance of insulin secretion and IR, one of the effects may dominate over the other [123,124]. Dauriz et al. [125] conducted a study on patients with newly diagnosed diabetes and found a positive effect of IL-6 on β-cell function, which was independent of weight loss. Moreover, a potentially similar effect was observed when it comes to IL-10, but this may be caused by the fact that IL-6 and IL-10 levels are coregulated [125]. In comparison, another 2-year study did not reveal an association between IL-6, IL-18, and glycated hemoglobin (HbA1c) levels [126]. Further studies are needed on the associations of IL-6 and IL-10 with pancreatic function to prove or disprove this relationship.

A link between inflammation and insulin secretion by the pancreas has been shown. Reducing the concentration of hs-CRP results in an improvement in this process and reduces the risk of developing DM2. Importantly, the change in hs-CRP level was also associated with weight loss [127]. In a 2-year study, it has been shown that rising hs-CRP, soluble E-selectin (sE-selectin), and soluble ICAM-1 (sICAM-1) levels result in the elevation of the HbA1c parameter in DM2 subjects. Increased levels of sE-selectin were also associated with impairment of insulin secretion in response to glucose [126]. These data are in line with the findings of another study indicating that obesity results in increased levels of inflammatory markers (e.g., IL-6, IL-8, IL-33, IL-1, IL-2, and TNF-α). Interestingly, obese patients with DM2 showed slightly lower levels of inflammatory markers than those with obesity without DM2. Obesity itself is a factor that drives changes in inflammatory markers. It is worth emphasizing that even despite weight loss, elevated inflammatory parameters persist in patients with type 2 diabetes, which proves a more complex mechanism of their action [128]. hs-CRP, CAM-1, and vascular cell adhesion protein 1 (VCAM-1) are examples of inflammatory markers that correlate with body fat percentage [129]. On the other hand, Dauriz et al. [125] conducted a study on subjects with newly diagnosed DM2, and no association was observed between the CRP and TNF-α levels with the β-cell state [125]. This study is in line with the findings of Dominguez et al. [130], which showed that among obese subjects with DM2 diagnosis, TNF-α blocker therapy (etanercept) resulted in a significant reduction in IL-6 and CRP levels but did not result in improvement in β-cell function or insulin sensitivity [130]. On the other hand, a study showed that the infusion of TNF-α in healthy subjects resulted in a decrease in basal insulin levels. This TNF-α action could be caused by an increase in insulin transfer in fatty tissue and muscles, deterioration of pancreatic insulin secretion, and escalated hepatic and kidney clearance of insulin [131].

Taking into account the importance of the inflammatory process in the pathogenesis and development of DM2, and the standards of diabetes treatment that rather focus on controlling serum glucose levels, research on maintaining or restoring β-cell function is justified to introduce new standards of treatment [116]. The pathogenesis of DM2 in terms of inflammation is presented in Figure 2.

## 5. Inflammation as a Link between NAFLD and DM2

The role of the inflammatory process in the pathogenesis of DM2 and NAFLD was highlighted in previous sections; nevertheless, the interplay between inflammation, DM2, and NAFLD should be discussed in more detail.

It is known that the *PPARGC1A* gene, which encodes the PGC-1α protein, has a pivotal role in mitochondria functions and also affects antioxidant management and inflammatory processes. Its expression is affected by many factors, including physical activity [132]. Variants of *PPARGC1A* have been associated with the risk of DM2 [133]. An alteration in a regulatory pathway in muscles from patients with DM2, which controls the expression of PGC-1α and mitofusin-2 induction, was revealed. In DM2, the mRNA expression of this protein in skeletal muscles is lower than that in healthy subjects [134,135]. PGC-1α protein is also related to NAFLD progression through oxidative stress, mitochondrial functions, gluconeogenesis, and lipogenesis management [94]. It was revealed that the upregulation of the FGF21–PGC-1α pathway results in an improvement in liver tissue in NAFLD subjects [136]. PGC-1α activity is regulated by sirtuin 1 (SIRT1). The SIRT1 function is decreased in subjects with DM2, and this leads to a reduction in PGC1-α activeness in the liver, an increase in oxidative stress and mitochondrial dysfunction, and liver damage, followed by NAFLD development [137]. Increased fatty acid oxidation and restraint of de novo lipogenesis are regulated by the SIRT1–AMPK pathway. A rat model revealed that SIRT1–liver kinase B1–AMPK regulation may alleviate lipid-associated liver disorders [138].

The IFN-γ-induced protein (IP-10), functionally categorized as a proinflammatory chemokine, was proposed as a diagnostic biomarker for NAFLD and DM2. Correlations between IP-10 and insulin resistance were also found [139]. Barchetta et al. [140] focused on the procollagen-III peptide (PIIINP) in terms of DM2. PIIINP is a marker of fibrosis and is therefore mainly associated with NAFLD progression to NASH. Their study findings suggest that, in the presence of DM2 and obesity, serum PIIINP levels are mostly associated with a systemic proinflammatory profile. This is probably related to adipose tissue expansion rather than fat deposition into the hepatic parenchyma [140]. 

It is known that obesity is associated with many serious complications, and the pathogenesis of NAFLD and DM2 is no different. The fat tissue secretion of inflammatory markers such as IL-6 may lead to the exacerbation of NAFLD and DM2 [141]. In one study conducted on obese participants, it was revealed that the inflammation of adipose tissue affected 44% of subjects. Moreover, adipose tissue inflammation is related to hyperinsulinemia, an increase in hepatic fat, visceral adipose tissue, and a loss of pancreatic β-cell function. The inflammation of adipose tissue may be a result of macrophage infiltration into disrupted adipocytes as a consequence of its extensive growth [142]. It is known that reducing body weight results in an improvement in insulin resistance and pancreatic β-cell function, as well as a decline in the amount of intrahepatic fat and a decrease in the expression of profibrotic agents in adipose tissue [143]. As a result of an increase in the amount of visceral adipose tissue, lipotoxicity is known to be a pathogenetic factor for the development of NAFLD and DM2. It is also responsible for decreasing lipid disposal mechanisms [144,145]. Lipotoxicity is known to be a factor responsible for liver necroinflammation and the subsequent destruction of the liver [146]. The impairment of insulin secretion was also observed as a result of β-cell lipotoxicity [147].

As insulin resistance is associated with chronic inflammation, it has been established that IR leads to the pathogenesis of DM2 and NAFLD [30]. In DM2, there is a growing tendency to develop IR, which is known to be a factor in increasing de novo lipogenesis in the liver, thus leading to NAFLD progression [148]. The role of TGF-β1 in NAFLD pathogenesis has been established, and this factor also plays a role in DM2 progression. TGF-1 increases the chance of developing DM2 by suppressing the creation of proteins required for the cells to function properly. The activation of Smad3, which binds to the insulin promoter and inhibits transcription, has been postulated as the mechanism through which TGF-1 targets insulin [149].

It was revealed that alteration in the gut microbiota leads to chronic inflammation in both DM2 and liver diseases, including NAFLD, which is associated with damage to the intestinal barrier [150]. Positive outcomes were observed as a result of probiotic and symbiotic supplementation when it came to inflammatory parameters [151], which is due to an alternating intestinal barrier. When the intestinal barrier is damaged, inflammatory markers, as well as gut microbiota and its metabolites, enter the general circulation and exacerbate the pathogenesis process of other diseases such as NAFLD [15,152]. The intestinal barrier dysfunction is associated with indigent glycemic control in subjects with DM2 [153]. It was also revealed that systemic inflammation results in increased intestinal permeability [154]. In summary, intestinal barrier dysfunction may be caused by inflammatory processes and poor glycemic control and result in NAFLD development. The relationships between NAFLD, DM2, and inflammation are presented in Figure 3.

## 6. The Outlook of Anti-Inflammatory Treatment in NAFLD and DM2

Due to the role of inflammatory processes in the development and progression of NAFLD and DM2, researchers more closely investigated these processes as a potential point of treatment. The common denominator in NAFLD and DM2 anti-inflammatory therapy may be aspirin, for which beneficial outcomes in both diseases were observed [155,156]. Aspirin belongs to the group of non-steroidal anti-inflammatory drugs (NSAIDs), which is one of the most commonly used medications around the world, with its application in painkillers, as well as anti-inflammatory and antipyretic agents [157]. In the context of NAFLD, a study conducted on mice revealed that aspirin treatment has preventive effects in terms of weight gain, glucose intolerance, and liver lipid accumulation in female subjects. However, no change was observed in the male group [158]. In our analysis of human studies, a reduction in the risk of NAFLD development and a decrease in liver fibrosis indexes due to follow-up treatments using aspirin can be highlighted [155,159]. The decrease in proteinuria in DM2 subjects was observed due to treatment with aspirin in combination with dipyridamole, which may indicate its potential nephroprotective effects in DM2 patients with nephropathy. Another study conducted on DM2 patients revealed that among females, the risk of dementia may be reduced [156,160]. It was shown that follow-up treatment of NSAID reduces the risk of DM2 development [161].

As in the pathogenesis of NAFLD, the importance of M1 and M2 Kupfer cells has been shown, and therefore researchers have sought to investigate their functions to treat this disease. Oxy210—a semi-synthetic oxysterol—is a substance that inhibits macrophage polarization. The amelioration of NASH and a decrease in white adipose tissue inflammation and levels of inflammatory cytokines in the liver and blood are also the results of Oxy210 use [71].

In DM2 individuals treated with hemodialysis, the anti-inflammatory and antidiabetic effects of linagliptin, one of the dipeptidyl peptidase-4 (DPP-4) inhibitors, and monotherapy outcomes were observed. When compared to pretreatment values, the amounts of prostaglandin E2 (PGE2), IL-6, and glycated albumin (GA) considerably dropped when linagliptin treatment began, while the amounts of GLP-1 dramatically increased [162]. Studies were conducted on the influence of GLP-1 on NAFLD and revealed that decreased GLP-1 degradation via the inhibition of DPP-4 results in anti-inflammatory and antifibrotic effects in the liver. Researchers found that treatment with evogliptin, which belongs to DPP-4 inhibitors, leads to the inhibition of releasing TGF-β and inducible nitric oxide synthase by Kupfer cells. This results in a decrease in HSC activity [163]. Treatment using another DPP-4 inhibitor, saxagliptin, resulted in an improvement in polarization from M1 to M2 macrophages and a decrease in the expression of NF-κB and TNF-α, thus highlighting the anti-inflammatory effects of this substance [73]. In a rodent model, another GLP-1 receptor agonist, exendin-4, reduced Mcp-1 expression and macrophage infiltration in the liver [164].

Due to the association of major metabolic components with lipid metabolism, medications improving these parameters are commonly used. Apart from their favorable effect on lipid profile parameters, these types of drugs are used to decrease inflammation. Krysiak et al. [165] assessed the effect of simvastatin and fenofibrate on inflammatory parameters. Their study revealed that plasma levels of hs-CRP, the monocyte release of TNF-α, interleukin-1, interleukin-6, MCP-1, lymphocyte release of interleukin-2, interferon, and TNF-α were all suppressed via simvastatin administration. hs-CRP, as well as monocyte and lymphocyte cytokine release, all decreased in response to fenofibrate medication [165].

Metformin is a first-line pharmacological agent for treating DM2. Metformin’s regulation of mitochondrial respiratory complex activity and homeostatic processes such as autophagy and mitophagy prevent the generation of cellular and mitochondrial reactive oxygen species (ROS) and the release of mitochondrial DNA, resulting in improved cellular health and lower proinflammatory cytokine production (such as IL-6, TNF-α, and IL1-β) [166]. Metformin also influences inflammation by regulating the Th17/Treg balance [166]. In lipopolysaccharide-activated macrophages, metformin inhibited the production of the proform of IL-1β, while it boosted the induction of the anti-inflammatory cytokine IL-10 [167,168]. Interestingly, the use of metformin was associated with decreased inflammation and hepatocyte proliferation in a zebrafish larvae model with NASH-associated hepatocellular carcinoma [169]. In another study, Mitrovic et al. aimed to evaluate the changes in inflammatory markers, fatty liver index (FLI), and fibrosis-4 (FIB-4) scores in non-obese metformin-treated DM2 patients with NAFLD. Their study showed a significant correlation between the levels of ferritin and C-reactive protein (CRP) and the FLI. The authors concluded that serum inflammatory markers with average normal values point to the efficacy of metformin monotherapy for inflammation control in non-obese DM2 patients with NAFLD [170].

La Grotta et al. [171] investigated the effect of SGLT-2 (sodium–glucose cotransporter 2) inhibitors on low-grade inflammation in patients with DM2. They showed that patients treated with SGLT-2 inhibitors had lower circulating levels of IL-6, uric acid, and fasting insulin than patients treated with other glucose-lowering drugs. The authors suggested that the anti-inflammatory effects of these drugs may be mediated by their ability to lower uric acid and insulin concentrations [171]. A mouse model study revealed that empagliflozin treatment reduces the infiltration of inflammatory cells in the pancreas and deregulates the pyroptosis-related inflammasome pathway by reducing NLRP3/caspase-1/Gasdermin D (GSDMD) expression. Thus, the protective effect of empagliflozin on pancreatic tissues was revealed [172]. Another SGLT-2 inhibitor is ipragliflozin, for which a beneficial effect on liver fibrosis was observed in subjects with DM2 [173].

A study using a rodent model revealed the efficacy of modafinil treatment in ameliorating NAFLD states. Modafinil is a substance that acts through cAMP alleviation in osteoblasts and fibroblasts. As cAMP is involved in the inflammatory process, it was revealed that modafinil may have anti-inflammatory and antifibrotic effects. The cited study revealed that these effects of modafinil are related to a decrease in K_Ca_2.3- and K_Ca_3.1-mediated signaling. The researchers proposed modafinil to treat liver diseases. The overexpression levels of K_Ca_2.3 and K_Ca_3.1 in murine models of liver disease and human-origin models treated with TGF were inhibited by modafinil. These findings indicate the antifibrotic action of MF in a mouse model of NASH and highlight the significance of K_Ca_2.3 and K_Ca_3.1 in the development of NASH [78]. 

Another drug that showed an effect in DM2-diagnosed patients with NAFLD treatment was diacerein. Diacerein is a drug mainly used in osteoarthritis treatment. It exhibits an anti-inflammatory effect by acting on IL-1 pathways. Diacerein reduced liver fibrosis compared with the placebo group. However, while diacerein is an anti-inflammatory drug in diabetic patients with NAFLD, it had no effect on cytokine levels compared with placebo [174].

In a study that aimed to block IL-1β with a human-engineered monoclonal antibody (gevokizumab) in patients with DM,2 it was observed that levels of IL-6 and TNF-α were significantly decreased. Moreover, after treatment, C-peptide showed an increased level, and the normalization of HbA1c was observed [175]. Noe et al. [113] revealed a statistically significant but clinically irrelevant decrease in HbA1c levels as a result of IL-1β antibody treatment (canakinumab). HbA1c is an indicator of the metabolic control of diabetes, and its values correlate with the mean levels of glycemia [113]. The connection between IL-1β and β cells was also shown in another study for treatment with an IL-1 receptor antagonist (anakinra) in subjects with impaired glucose tolerance. At the endpoint, due to this intervention, improvements were observed in first-phase insulin secretion and the insulinogenic index, which point to the enhancement of β-cell function [114]. Ruscitti et al. [176] revealed that among DM2 patients with rheumatoid arthritis, anakinra treatment leads to improvements in both diseases. At the endpoint, a significant reduction in HbA1c levels in the intervention group was observed, which indicates an enhancement in maintaining glucose levels. This study compared anakinra with TNF inhibitors, and improvement in HbA1c did not reach significance when using TNF inhibitors [176]. Similar effects to the effect of anakinra on the DM2 state were observed in treatment with sarilumab—an IL-6 receptor antagonist—in patients with DM2 and rheumatoid arthritis. Those patients reached significantly lower HbA1c levels than patients treated with a placebo and the group treated with TNF inhibitors [177]. Another study attempted to demonstrate the clinical effect of IL-1β antibody in DM2 treatment; however, no significant changes in C-peptide and HbA1c were found [116]. Thus, further studies on IL-1β inhibition are needed to prove or disprove its beneficial effects in DM2. The summary of described in this review animal models is performed in Table 1.

## 7. Conclusions

Summarizing the information presented here, the importance of inflammatory processes in the pathogenesis of NAFLD and DM2 should be emphasized. In light of our review, the relationships between NAFLD and DM2 are complex, and a better understanding of them requires further research, including studies on the inflammatory molecular processes of gene upregulation and downregulation, the expression of corresponding peptides, gene methylation patterns, and altered pathway signaling. Moreover, environmental factors undoubtedly play a key role in the pathogenesis of these diseases. Therefore, treatment aimed at the inflammatory process may be part of the management of both diseases. It is worth emphasizing that NAFLD and DM2 often occur together, one inducing the other and vice versa; therefore, the implementation of prevention methods will work in both cases. Fortunately, many drugs targeting inflammatory pathways are already being tested in clinical trials and, although there are currently no guidelines stating which medication works better in different clinical scenarios, there is hope for better management of NASH, DM2, and transitional states, namely NAFLD and insulin resistance. Undoubtedly, the best current solution is lifestyle change, which has been proven to influence many pathological pathways.

## Figures and Tables

**Figure 1 ijms-24-09677-f001:**
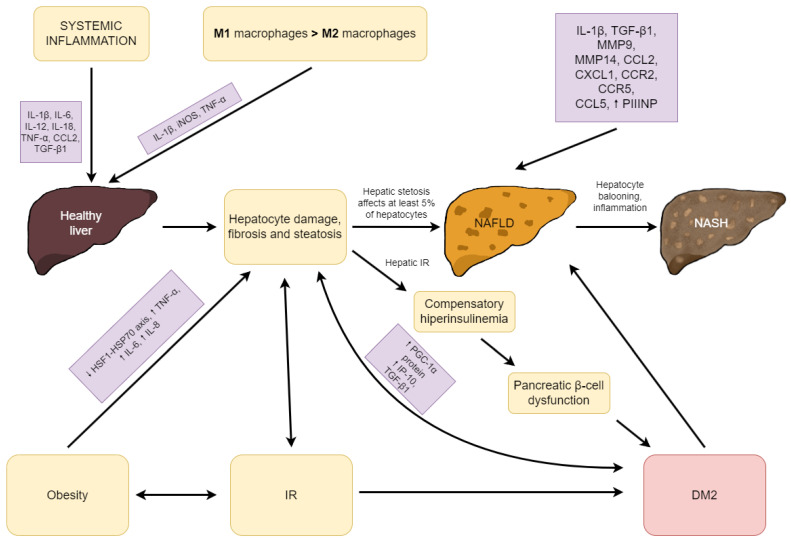
The pathogenesis of NAFLD and NASH in terms of inflammation. Abbreviations: CCL, chemokine (C–C motif) ligand; CCR, C–C chemokine receptors; CXCL1, chemokine (C–X–C motif) ligand 1; DM2, type 2 diabetes mellitus; HSF1, heat-shock transcription factor-1; HSP70, 70 kDa family of heat-shock proteins; IL, interleukin; iNOS, inducible nitric oxide synthase; IP-10, IFN-γ-induced protein (IP-10); IR, insulin resistance; MMP, matrix metalloproteinase; NAFLD, non-alcoholic fatty liver disease; NASH, non-alcoholic steatohepatitis; PIIINP, procollagen-III peptide; PGC-1α, peroxisome proliferator-activated receptor-γ coactivator 1α; TGF-β1, transforming growth factor beta 1; TNF-α, tumor necrosis factor α; ↑, increase; ↓, decrease.

**Figure 2 ijms-24-09677-f002:**
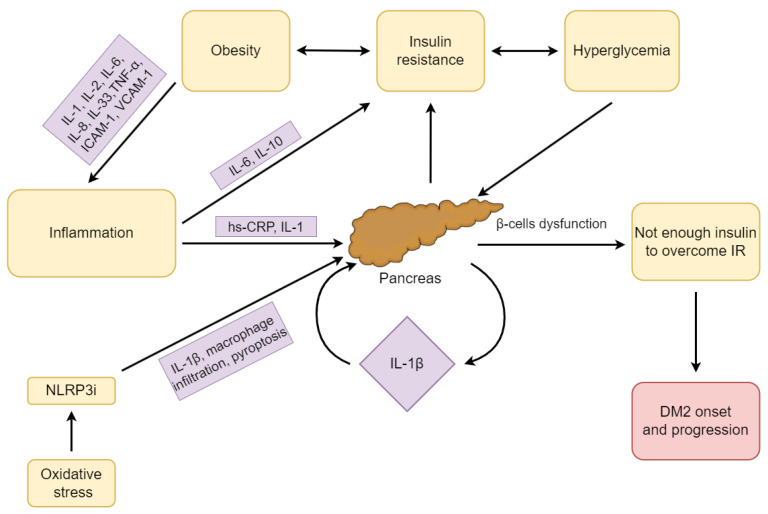
The pathogenesis of DM2 in terms of inflammation. Abbreviations: DM2, type 2 diabetes mellitus; hs-CRP, high-sensitivity C-reactive protein; ICAM-1, intercellular adhesion molecule-1; IL, interleukin; IR, insulin resistance; NLPR3i, domain-like receptor protein 3 (NLRP3) inflammasome; TNF-α, tumor necrosis factor α; VCAM-1, vascular cell adhesion molecule-1.

**Figure 3 ijms-24-09677-f003:**
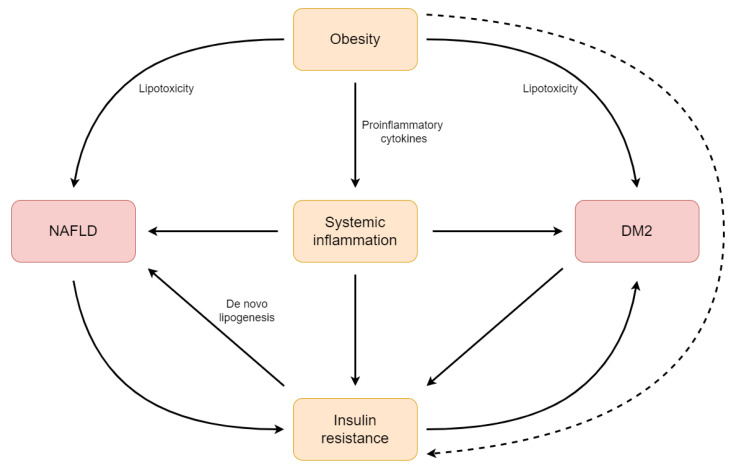
The relationships between NAFLD, DM2, and inflammation. Abbreviations: DM2, type 2 diabetes mellitus; NAFLD, non-alcoholic fatty liver disease.

**Table 1 ijms-24-09677-t001:** Selected studies on animal models and their outcomes in terms of inflammation, DM2, and NAFLD.

Study	Group	Inflammation, NAFLD, DM2 Induction	Intervention	Duration	Outcomes
Choi et al. [78]	110 C57BL/6 mice	Steatosis, steatohepatitis, and liver fibrosis induced by choline-deficient, L-amino-acid-defined, high-fat diet with 0.1% methionine	Modafinil in doses 10, 50, or 100 mg/kg	20 weeks	Used diet and TAA lead to K_Ca_2.3, K_Ca_3.1, upregulation, and downregulation of catalase in liver tissues Modafinil can reverse K_Ca_2.3, K_Ca_3.1, collagen, and α-smooth muscle actin upregulation and downregulation of catalase, and leads to a decline in the inflammatory response, collagen deposition, and α-smooth muscle actin expression
	75 C57BL/6 male mice	Hepatitis and fibrosis induced by TAA in dose 100 mg/kg intraperitoneally 3 times per week	Modafinil in doses 10, 50, or 100 mg/kg	16 weeks	
Zhao et al. [80]	50 male Wistar rats	Liver fibrosis induced by carbon tetrachloride CCl4 subcutaneous injections and high-lipid/low-protein diet for 8 weeks	DHC administration in doses 0.5 or 1.0 g/kg	8 weeks	TGF-β1 and the expression of Gremlin mRNA and protein were higher than in the control group Expression of BMP-7 mRNA and protein lower than in the control group Improvement in liver fibrosis, Decrease in TGF-β1 and the mRNA and protein expression of Gremlin in DHC groups
Weng et al. [81]	483 Sprague Dawley rats (205 in the CCL4 group and 278 in the DMN group)	Liver fibrosis induced by subcutaneous injection of CCl4 or intraperitoneal injection of DMN	IFN-γ administration (in different doses: 1.67 MU/kg daily, 5 MU/kg daily, and 15 MU/kg daily)	8 weeks for the CCL4 group 4 weeks for the DMN group	IFN-γ administration decreased the HSCs activation and is effective in reducing liver fibrosis The results of IFN-γ are dose-dependent (better results are achieved with a higher dose)
Tang et al. [91]	40 male C57BL/6 mice	6-week methionine choline-deficient diet to establish NASH	2 weeks administration of CTSB inhibitor (CA-074 methyl ester)	6 weeks	Higher expression of CTSB and caspase-1 than in the normal diet group Possible regulation of caspase-1 levels by CTSB CTSB inhibition results in a decline in IL-1β and IL-18 levels and downregulation of NLRP3 inflammasome in KCs
Benrick et al. [108]	18 IL-6^−/−^ mice and 18 wild-type mice	High-fat diet to induce weight gain	Access to running wheels and lack of this access	4 weeks	IL-6 contributes to the exercise-associated increase in insulin sensitivity A high-fat diet without running led to impairing insulin sensitivity; in contrast, running was a preventive factor in conditions of insulin sensitivity in wild-type but not in IL-6^−/−^ mice
Huang et al. [117]	C57BL/6 mice and a mice insulinoma immortalized β-cell line MIN6	Diabetes induced by streptozotocin in dose 40 mg/kg intraperitoneal for five days or high-fat and high-sucrose diet	AAV8 to induce expression of Kindlin-2	12 weeks	Insufficiency of Kindlin-2 leads to exacerbating diabetes, promotes β-cell inflammation and dysfunction induced by a high-fat diet Overexpression of Kindlin-2 improves insulin secretion and ameliorates diabetes induced by streptozotocin In vitro model of high-glucose-induced β-cell dysfunction revealed that overexpression of Kindlin-2 leads to decreased expression of proinflammatory cytokines and NLRP3 inflammasome in β cells
Jiang et al. [120]	Mouse islet β-TC-6 cells		PCB118 (5, 10, and 20 nmol/L)	48 or 72 h	NLRP3 inflammasome signaling pathways in β cells are important in diabetes development
Abderrazak et al. [121]	12 ApoE_2_.Ki female mice	Chronic high-fat diet	Arglabin 2.5 ng/g twice a day in intraperitoneal injection	13 weeks	Inhibition of NLRP3 caused by arglabin leads to a decrease in inflammation and apoptosis in pancreatic β cells
Yang et al. [122]	24 diabetes-prone C57BLKS/J-Leprdb/Leprdb (db/db) male mice and 24 wild-type male mice		WMW in different doses (4800, 9600, and 19,200 mg/kg)	4 weeks	Compared with the control group diabetic mice had higher protein expression levels of NLRP3 inflammasome components NLRP3 and caspase-1 (P20) than wild-type mice WMW decreases caspase-12, increases Bcl-2 expression, and decreases the upregulated production of IL-1β,IL-18,MCP-1α, and macrophage-specific surface glycoprotein F4/80 in diabetic mice
Sharma et al. [138]	36 male Wistar rats	NAFLD induced by 12 weeks high-fat diet	Berbamine (50 or 150 mg per kg)	12 weeks + 28 days	Improvements in liver function, liver index, and liver image due to berbamine administration Inducing the SIRT1/LKB1/AMPK pathway leads to protection against hepatic lipid metabolic disorders
Zhou et al. [158]	Mice with maternal overnutrition	Obesity and NAFLD induced by high-fat diet + diethylnitrosamine (intraperitoneally 20–25 μg/g and 50 μg/L in drinking water at age 21 days)	90 μg to 120 μg aspirin per day	12 weeks	Improvement in insulin/Akt signaling, activation of AMPK signaling, inhibition of Wnt-signaling and MAPK signaling leads to improvements in glucose intolerance, weight gain, and liver fat accumulation in female mice

Abbreviations: AMPK, 5′-adenosine monophosphate (AMP)-activated protein kinase; Akt, protein kinase B; Bcl-2, B-cell leukemia 2; BMP-7, bone morphogenetic protein-7; CCL4, carbon tetrachloride; CTSB, Cathepsin B; DHC, Danshao Huaxian capsule, DMN, dimethylnitrosamine; HSCs, hepatic stellate cells; IL, interleukin; KCs, Kupfer cells; MAPK, mitogen-activated protein kinases; MCP-1α, monocyte chemoattractant protein-1α; NLRP3, NOD-like receptor protein 3; SIRT1, sirtuin 1; LKB1, liver kinase B1; TAA, thioacetamide; TGF-β1, transforming growth factor beta 1; WMW, Wu–Mei–Wan.

## Data Availability

Not applicable.

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
