# Peer review of "Type 2 Diabetes Mellitus, Non-Alcoholic Fatty Liver Disease, and Metabolic Repercussions: The Vicious Cycle and Its Interplay with Inflammation"

_ijms, 2023, doi:10.3390/ijms24119677_

Round 1
Reviewer 1 Report
the manuscript by Rafat Frankowski., et al., titled as “Inflammation as a link between non-alcoholic fatty liver disease and diabetes mellitus type 2” is a systematic review manuscript of 152 research article data extensively reviewed by authors, and presented in easily understandable knowledge source.
Here are my comments
1. the manuscript written well and reviwed all the source content in good anaytical way possible, I would ask authors to draw the liver/hepatocyte illustrations and present the signalling cascade in more scientific way than this present images.
You may use biorender/adobe illustrator for this, it will greatly improve the understanding of the review. Specially for early stage researchers.
2. authors could include a table describing the animal models and in-vitro/ex-vivo models, chemical agents/lignads (to induce inflammation) used in all the reviwed content, it will help to understand, what methods are usefull to study inflammation in liver NASH/DM2, it will serve as a one-stop knowledge source.
English language used in the manuscript is good, minor spelling mistake may have to be checked in proof-read.
Author Response
Response to Reviewer 1 Comments
Thank the Reviewer’s very much for your time, profound analysis and valuable comments on our manuscript. The responses for all points are below. The changes were introduced into the text of manuscript, as suggested by the Reviewer.
Point 1: the manuscript written well and reviewed all the source content in good analytical way possible, I would ask authors to draw the liver/hepatocyte illustrations and present the signaling cascade in more scientific way than this present images
Response 1: Thank the Reviewer’s suggestion. In our opinion, the presented diagrams clearly show the influence of inflammation on the pathogenesis of NAFLD, and the addition of further figures will duplicate the information contained in the text. However, we decided to include the figures in the diagrams for better clarity in lines 1161 and 1271.
Point 2: authors could include a table describing the animal models and in-vitro/ex-vivo models, chemical agents/lignads (to induce inflammation) used in all the reviewed content, it will help to understand, what methods are usefull to study inflammation in liver NASH/DM2, it will serve as a one-stop knowledge source.
Response 2: Thank the Reviewer’s suggestion. We included a table describing animal models analyzed in our review as the Reviewer requested in line 1489.
We sincerely hope that all changes introduced by us in the text will be fully satisfactory for the Reviewer.
Reviewer 2 Report
The work is well structured and correct. I had only initial doubts since I could not see the additional methods and tables that characterize the systematic review.
Although there are articles with a similar theme in the literature (Dharmalingam M, Yamasandhi PG. Nonalcoholic Fatty Liver Disease and Type 2 Diabetes Mellitus. Indian J Endocrinol Metab. 2018 May-Jun;22(3):421-428. doi: 10.4103/ijem.IJEM_585_17.), a systematic revision of the literature has never been made about it: this represents the point of novelty.
The references are relevant.
Please use the same format and font in figure 1 and 2.
Line 105: put the definition of the acronym IL, that is interleukin, and not IL-6, otherwise you should specify it for each interleukin.
Line 534: Please specify the acronym GA and PGE2 in full
Line 390/535: glucagon-like peptide 1 (GLP-1): only the first time the acronym appears in the manuscript must be inserted in full.
Best regards.
Author Response
Response to Reviewer 2 Comments
Thank the Reviewer’s very much for your time, profound analysis and valuable comments on our manuscript. The responses for all points are below. The changes were introduced into the text of manuscript, as suggested by the Reviewer.
Point 1: Please use the same format and font in figure 1 and 2.
Response 1: Thank the Reviewer’s suggestion. We changed the font format in the figures so that it was the same in each one.
Point 2: Line 105: put the definition of the acronym IL, that is interleukin, and not IL-6, otherwise you should specify it for each interleukin.
Response 2: Thank the Reviewer’s suggestion. We defined the acronym as the Reviewer requested in line 393.
Point 3: Line 534: Please specify the acronym GA and PGE2 in full
Response 3: Thank the Reviewer’s suggestion. We specified the acronyms as the Reviewer requested in line 1390.
Point 4: Line 390/535: glucagon-like peptide 1 (GLP-1): only the first time the acronym appears in the manuscript must be inserted in full.
Response 4: Thank the Reviewer’s suggestion. We defined the acronym as the Reviewer requested only in the first time that it appears.
We sincerely hope that all changes introduced by us in the text will be fully satisfactory for the Reviewer.
Reviewer 3 Report
The review presented by Frankowsky et al. is very interesting and necessary, and the approach taken is a great exercise that many scientists in the field would appreciate. The three figures are very good, clear, and useful. However, although I believe the general structure is alright, there are obvious differences between different parts, most probably written by different co-authors. The different sections are not even; some of them are not well written and I think a senior co-author should review the whole text to homogenize the manuscript.
As an example, I would humbly suggest how I would change the conclusion section based on the text presented (I also attach the text with tracked changes).
Conclusions
Summarizing the information presented here, the importance of inflammatory processes in pathogenesis of NAFLD and DM2 should be emphasized. In the light of our review the relationships between NAFLD and DM2 are complex and a better understanding of them requires further research including the inflammatory molecular processes of gene upregulation and downregulation, expression of corresponding peptides, gene methylation patterns and altered pathway signaling. Moreover, environmental factors undoubtedly play a key role in the pathogenesis of these diseases. Therefore, treatment aimed at the inflammatory process may be part of the management of both states. It is worth emphasizing that these entities often occur together, one inducing the other and vice versa, and therefore, the implementation of prevention methods will work in both cases. Fortunately, many drugs targeting inflammatory pathways are already being tested in clinical trials and, although there are currently no guidelines stating which medication works better in different clinical scenarios, there is hope for better management of NASH, DM2 and transitional states: NAFLD, insulin resistance. Beyond doubt, currently the best solution is lifestyle modification that has proven affect many pathological pathways.

There is also a serious need to review the English language throughout the entire manuscript. The phrasing is sometimes complicated and repetitive making difficult to understand the conclusions emerging from their findings. I would suggest a whole language editing by for example the IJMS Service or others.
Author Response
Response to Reviewer 3 Comments
Thank the Reviewer’s very much for your time, profound analysis and valuable comments on our manuscript. The responses for all points are below. The changes were introduced into the text of manuscript, as suggested by the Reviewer.
Point 1: I would humbly suggest how I would change the conclusion section based on the text presented (I also attach the text with tracked changes).
Conclusions: Summarizing the information presented here, the importance of inflammatory processes in pathogenesis of NAFLD and DM2 should be emphasized. In the light of our review the relationships between NAFLD and DM2 are complex and a better understanding of them requires further research including the inflammatory molecular processes of gene upregulation and downregulation, expression of corresponding peptides, gene methylation patterns and altered pathway signaling. Moreover, environmental factors undoubtedly play a key role in the pathogenesis of these diseases. Therefore, treatment aimed at the inflammatory process may be part of the management of both states. It is worth emphasizing that these entities often occur together, one inducing the other and vice versa, and therefore, the implementation of prevention methods will work in both cases. Fortunately, many drugs targeting inflammatory pathways are already being tested in clinical trials and, although there are currently no guidelines stating which medication works better in different clinical scenarios, there is hope for better management of NASH, DM2 and transitional states: NAFLD, insulin resistance. Beyond doubt, currently the best solution is lifestyle modification that has proven affect many pathological pathways.
Response 1: Thank the Reviewer’s suggestion. We reconsidered the phrase as the Reviewer requested in lines 1501-1516.
Point 2: There is also a serious need to review the English language throughout the entire manuscript
Response 2: Thank the Reviewer’s suggestion. We reviewed the English language in the manuscript.
We sincerely hope that all changes introduced by us in the text will be fully satisfactory for the Reviewer.
Reviewer 4 Report
Dear editor of IJMS (MDPI) and authors of the manuscript ijms-2394791, I am pleased to accept this review invitation and take place in the peer review process of this study.
I have finalized the revisions for the whole manuscript, initially providing seven corrections pages and making them accessible to the authors. Upon the authors' integration of these initial changes, I will offer additional comments for the remaining text sections. By adopting this approach, which accounts for numerous corrections, I am confident it will significantly enhance the overall revision process.
Please, dear authors, resolve all the commentaries of the annexed WORD document. These commentaries include the text structure, English editing, and methodology suggestions. When you finish these first comments, I will provide more analyses.
With best regards,
The Reviewer.

English style needs extensive review and is incomprehensible in many text parts. I think asking the authors to acquire an English correction service like PaperTrue or MDPI may be necessary.
Author Response
Response to Reviewer 4 Comments
Thank the Reviewer’s very much for your time, profound analysis and valuable comments on our manuscript. The responses for all points are below. The changes were introduced into the text of manuscript, as suggested by the Reviewer.
Point 1: Please, consider ''The Role of Inflammation in Connecting Non-Alcoholic Fatty Liver Disease and Type 2 Diabetes Mellitus'' or create a new one with the word ''interplay'' like ''Type 2 Diabetes Mellitus, Non-Alcoholic Fatty Liver Disease and Metabolic Repercussions: The Vicious Cycle and the Interplay with Inflammation''. This last is better.
Response 1: Thank the Reviewer’s suggestion. According to it, we changed the manuscript title to''Type 2 Diabetes Mellitus, Non-Alcoholic Fatty Liver Disease and Metabolic Repercussions: The Vicious Cycle and the Interplay with Inflammation”
Point 2: Is this correct? Please, confirm!
Response 2: The affiliation Students' Research Club, Department of Clinical Pharmacology, Medical University of Lodz, 90-153 Lodz, Poland is correct.
Point 3: Reconsider to ''The prevalence of metabolic-related disorders, such as non-alcoholic fatty liver disease (NAFLD) and type 2 diabetes mellitus (DM2), has been increasing. Therefore, developing improved methods for preventing, treating, and recognizing these two conditions is also necessary. In this study, our primary focus was on examining the role of chronic inflammation as a potential link in the pathogenesis of these diseases and their interconnections. A comprehensive search of the PubMed database using keywords such as "non- alcoholic fatty liver disease," "type 2 diabetes mellitus," "chronic inflammation," "pathogenesis," and "progression" yielded 152 relevant papers for our analysis. The findings of our study revealed intricate relationships between the pathogenesis of NAFLD and DM2, emphasizing the crucial role of inflammatory processes. These connections involve various molecular functions, including altered signaling pathways, patterns of gene methylation, expression of related peptides, and up-and-down-regulation of several genes. Our study is a foundational platform for future research into the intricate relationship between NAFLD and DM2, allowing for a better understanding of the underlying mechanisms and the potential for introducing new treatment standards.''
Response 3: Thank the Reviewer’s suggestion. We reconsidered the phrase as the Reviewer requested in lines 19-32.
Point 4: Please, separate your keywords with '';''.
Response 4: Thank the Reviewer’s suggestion. We separated our keywords with “;”.
Point 5: Also, add a dot ''.'' at the end of the keywords section.
Response 5: Thank the Reviewer’s suggestion. We added a dot at the end of the keywords section.
Point 6: Reconsider to ''NAFLD refers to a condition characterized by hepatic steatosis, which involves the accumulation of fat in at least 5% of hepatocytes in the liver. This condition occurs without any identifiable secondary causes such as alcohol use or hepatitis C virus infection.''
Response 6: Thank the Reviewer’s suggestion. We reconsidered the phrase as the Reviewer requested in lines 38-41.
Point 7: Reconsider to ''NAFLD has emerged as the leading liver disease in terms of prevalence, accounting for approximately 32.4% of cases and exhibiting a rising trend in recent years. In terms of gender distribution, NAFLD is more frequently diagnosed in men (39.7%) compared to women (25.6%).''
Response 7: Thank the Reviewer’s suggestion. We reconsidered the phrase as the Reviewer requested in lines 41-44.
Point 8: Reconsider to ''As NAFLD progresses, it may advance to a more severe condition known as non-alcoholic steatohepatitis (NASH). In NASH, additional components such as liver damage, inflammation, and fibrosis become evident, indicating a more pronounced level of liver injury.''
Response 8: Thank the Reviewer’s suggestion. We reconsidered the phrase as the Reviewer requested in lines 109-112.
Point 9: Reconsider to ''The diagnosis of NASH can be made based on several critical indicators, including the initial statement of at least 5% liver steatosis (fat accumulation in liver cells). Additionally, varying degrees of hepatic fibrosis (abnormal accumulation of fibrous tissue), hepatocyte ballooning degeneration (swelling and damage to liver cells), and lobular inflammation (inflammation in the lobules of the liver) are considered significant factors in confirming a diagnosis of NASH.''
Response 9: Thank the Reviewer’s suggestion. We reconsidered the phrase as the Reviewer requested in lines 112-117.
Point 10: In which region/country?
Response 10: Thank the Reviewer’s suggestion. We determined that this statement referred to the United States and we included this information in line 121.
Point 11: Avoid passive voice here.
Response 11: Thank the Reviewer’s suggestion. We rephrased sentence to avoid passive voice in lines117-118.
Point 12: Add a comma before this ''and''.
Response 12:T hank the Reviewer’s suggestion. We added a comma before “and” in line118.
Point 13: Reconsider to ''shortly''.
Response 13: Thank the Reviewer’s suggestion. We reconsidered the phrase as the Reviewer requested in line 118.
Point 14: Reconsider to ''NASH poses a significant health concern, impacting approximately 1.5-6.5% of the general population. It ranks among the primary causes of liver cirrhosis in the United States, highlighting the potential for severe liver damage as the disease progresses. Early detection and appropriate management of NASH are crucial to mitigate these risks.''
Response 14: Thank the Reviewer’s suggestion. We reconsidered the phrase as the Reviewer requested in lines 119-123.
Point 15: Reconsider to ''NAFLD, a metabolic disorder, connects with other metabolic defects such as obesity, dyslipidemia, insulin resistance, and diabetes mellitus (DM). Nevertheless, emphasizing that this phenomenon occurs even in individuals with average weight is crucial.''
Response 15: Thank the Reviewer’s suggestion. We reconsidered the phrase as the Reviewer requested in lines 124-126.
Point 16: Cite the underlying mechanisms.
Response 16: Thank the Reviewer’s suggestion. We cited underlying mechanisms and added references (3,11,12) in lines 125-131.
Point 17: Reconsider to ''The growing prevalence of the diseases above points to an anticipated rise in NAFLD cases. NAFLD, in turn, serves as a trigger, heightening the risk of developing other disorders. Notably, NASH, cirrhosis, and hepatocellular carcinoma take the forefront. NASH specifically predisposes individuals to cirrhosis in about 20% of cases and hepatocellular carcinoma, cardiovascular diseases, and other malignancies.''
Response 17: Thank the Reviewer’s suggestion. We reconsidered the phrase as the Reviewer requested in lines 131-136.
Point 18: Reconsider to ''In the literature, there is an extensive exploration of the connections between NAFLD and the risk of several cancers, such as colorectal, kidney, uterine, and bladder cancer. However, the highest risk is observed for hepatocellular carcinoma. Additionally, researchers have documented an elevated risk of developing diabetes mellitus, colonic diverticulosis, cardiovascular diseases, and kidney diseases among individuals with NAFLD.''
Response 18: Thank the Reviewer’s suggestion. We reconsidered the phrase as the Reviewer requested in lines 137-144.
Point 19: Please, for each mentioned disease, provide a specific reference right after the disease mention.
Response 19: Thank the Reviewer’s suggestion. We added a specific references (14-19) right after the diseases mention.
Point 20: Also, please provide possible mechanisms for the cited links between NAFLD and cancer development.
Response 20: Thank the Reviewer’s suggestion. We explained possible mechanisms for links between NAFLD and cancer development in lines 140-141.
Point 21: Reconsider to ''NAFLD is acknowledged to increase mortality in affected patients, primarily due to the underlying mechanisms rooted in atherosclerosis and cardiovascular diseases.''
Response 21: Thank the Reviewer’s suggestion. We reconsidered the phrase as the Reviewer requested in lines 144-147.
Point 22: Please, explain the underlying mechanisms that you cited point-by-point.
Response 22: Thank the Reviewer’s suggestion. We explained the underlying mechanisms that we cited in lines 146-147.
Point 23: Reconsider to ''A recent proposal suggests using metabolic dysfunction-associated fatty liver disease (MAFLD) instead of NAFLD, as it provides a broader perspective and emphasizes the connection between this condition and metabolic dysfunction. Clinicians can establish the diagnosis of MAFLD when liver steatosis coexists with at least one of the following criteria: overweight, diagnosis of type 2 diabetes (DM2), or the presence of at least two out of five metabolic syndrome risk factors.''
Response 23: Thank the Reviewer’s suggestion. We reconsidered the phrase as the Reviewer requested in lines 148-153.
Point 24: Please, ensure that all abbreviations throughout the text are described and used repeatedly after their first mention. Revise the whole manuscript for errors regarding abbreviations. In the following paragraph, ''Diabetes mellitus'' is not replaced by ''DM''.
Response 24: Thank the Reviewer’s suggestion. We revised the whole manuscript for errors regarding abbreviations.
Point 25: Reconsider ''DM comprises a group of metabolic diseases that exhibit chronic hyperglycemia. These conditions stem from defects in either insulin secretion, insulin action, or both.''
Response 25: Thank the Reviewer’s suggestion. We reconsidered the phrase as the Reviewer requested in lines 154-155.
Point 26: Reconsider to ''The development of DM2 arises from a gradual decline in β-cell insulin secretion, primarily occurring in the presence of insulin resistance and metabolic syndrome and without the involvement of autoimmune factors.''
Response 26: Thank the Reviewer’s suggestion. We reconsidered the phrase as the Reviewer requested in lines 160-162.
Point 27: Be specific with the DM types. Is this data from DM2? Please, rephrase and clarify.
Response 27: Thank the Reviewer’s suggestion. We rephrase and clarified this sentence in lines 158-160.
Point 28: Reconsider to ''It is crucial to note that the prevalence of diabetes mellitus (DM) is on the rise. According to projections, by 2045, approximately 783.2 million individuals, which accounts for 12.2% of adults aged 20-79 years, are expected to be living with diabetes.''
Response 28: Thank the Reviewer’s suggestion. We reconsidered the phrase as the Reviewer requested in lines 157-160.
Point 29: Be specific with the DM types. Please, rephrase for clarification.
Response 29: Thank the Reviewer’s suggestion. We rephrase and clarified this sentence in lines 157-160.
Point 30: Consider adding a ''higher'' between ''at'' and ''risk''.
Response 30: Thank the Reviewer’s suggestion. We added a “higher” between “at” and “risk” in line 352.
Point 31: This is incomprehensible and needs clarification with rephasing and a little explanation about the underlying mechanisms.
Response 31: Thank the Reviewer’s suggestion. We rephrased this paragraph and add a little explanation to the mechanisms in lines 353-357.
Point 32: Foreign organisms cause infections, and infections cause neither DM1 nor DM2. Additionally, autoimmune diseases are not associated with responses against foreign organisms but with self- antigens. Please, rephrase the whole paragraph for clarification.
Response 32: Thank the Reviewer’s suggestion. We rephrased this paragraph in lines 358-366.
Point 33: What process? General inflammatory responses? Inflammation against germs? Inflammation in autoimmune diseases? Inflammation against cancer? ... Can you understand how this paragraph confuses the reader?
Response 33: Thank the Reviewer’s suggestion. We rephrased this paragraph in lines 358-366.
Point 34: Reconsider to ''Additionally, there is a significant correlation between adipose tissue and low-grade inflammation, particularly in individuals who are obese. This relationship highlights the link between excess body fat and chronic, low-level inflammation.''
Response 34: Thank the Reviewer’s suggestion. We reconsidered the phrase as the Reviewer requested in lines 367-369.
Point 35: Reconsider to ''Studies have indeed demonstrated that in individuals with morbid obesity, pro-inflammatory genes are more expressed in subcutaneous adipose tissue than visceral adipose tissue. This observation suggests that the subcutaneous fat layer, which lies just beneath the skin, may significantly influence inflammation in cases of severe obesity.''
Response 35: Thank the Reviewer’s suggestion. We reconsidered the phrase as the Reviewer requested in lines 370-373.
Point 36: Revise abbreviations to avoid repetition of expressions, too.
Response 36: Thank the Reviewer’s suggestion. We revised the abbreviations.
Point 37: Reconsider to ''Indeed, research has provided evidence of a strong connection between visceral adipose tissue and inflammation. A notable study demonstrated decreased C-reactive protein (CRP) levels after omentectomy (surgical removal of the omentum) in individuals who underwent the Roux-en- Y gastric bypass procedure. This reduction in CRP levels following the removal of visceral adipose tissue underscores this type of fat's significant role in promoting inflammation.''
Response 37: Thank the Reviewer’s suggestion. We reconsidered the phrase as the Reviewer requested in lines 374-381.
Point 38: Provide more references here to prove your point of omentectomy reducing CRP. For me, this is only partially right and must be revised consistently. The present evidence level is not higher enough.
Response 38:Thank the Reviewer’s suggestion. We rephrased this sentence and described this CRP decline due to omentectomy as a possible but also we referred a study (reference 35) which didn’t prove this CRP decline after this procedure to show that it is still not entirely clear if omentectomy affect the inflammatory process.
Point 39: Consider adding a ''the'' between ''in'' and ''development''.
Response 39: Thank the Reviewer’s suggestion. We added a “the” between “in” and “development” in line 384.
Point 40: Consider adding a ''the'' between ''in'' and ''development''.
Response 40: Thank the Reviewer’s suggestion. We added a “the” between “in” and “development” in line 387.
Point 41: Rephrase for better understanding. These whole sentences are now incomprehensible.
Response 41: Thank the Reviewer’s suggestion. We rephrased requested sentences in lines 388-192.
Point 42: Consider adding a ''comma'' before ''and''.
Response 42: Thank the Reviewer’s suggestion. We added a “comma” before “and“in line 394.
Point 43: Consider ''considering
Response 43: Thank the Reviewer’s suggestion. We rephrased requested sentences in lines 397-400.
Point 44: Rephrase for better understanding. This whole paragraph is now incomprehensible.
Response 44: Thank the Reviewer’s suggestion. We rephrased requested sentences in lines 397-400.
Point 45: Please, exclude this whole paragraph and rewrite the methodology following ''2.1. Focal question'', ''2.2. Language'', ''2.3. Databases'', ''2.4. Study extraction'', ''2.5. Data extraction'', and ''2.6. Quality assessment''.
Response 45: Thank the Reviewer’s suggestion. We rephrased the whole “Materials and methods” paragraphas the Reviewer requested in lines 401-402 and 460-492.
Point 46: These are ''results'' and must be excluded. As a narrative review paper, I think it must not contain this paragraph.
Response 46: Thank the Reviewer’s suggestion. We excluded the “Results” paragraph.
Point 47: Please, follow MDPI's standard of manuscript templates. This is not correct.
Response 47: Thank the Reviewer’s suggestion, however we couldn’t find inaccuracies with MDPI’s standards of manuscript templates.
Point 48: I don't think epidemiological data predisposes this affirmation. Please, dear authors, reconsider this paragraph and add additional good information about the underlying mechanisms.
Response 48: Thank the Reviewer’s suggestion. We reconsidered this paragraph and included explanation of the underlying mechanisms in lines 498-505.
Point 49: It may not sound evident to the reader. Please, rephrase.
Response 49: Thank the Reviewer’s suggestion. We rephrased requested sentences in lines 508-509 and 587-588.
Point 50: These sentences may sound confusing to the reader and must not present the author’s opinion about the issue. Please, rephrase for better clarification and provide references that prove your point of view, avoiding e
Response 50: Thank the Reviewer’s suggestion. We rephrased requested sentences and added references 46,48,49 to prove our point of view in lines 508-509 and 587-588.
Point 51: Rephrase, improve punctuation, and provide each factor a reference right after each factor's first mention to improve the manuscript's quality and scientifically.
Response 51: Thank the Reviewer’s suggestion. We rephrased requested sentences and added references (50-55)right after each factor in lines 589-593.
Point 52: Reconsider to ''and is also followed''.
Response 52: Thank the Reviewer’s suggestion. We reconsidered the phrase as the Reviewer requested in line 595.
Point 53: IR of the liver, specifically? Please, clarify, as you cited right above ''hepatic IR''.
Response 53: Thank the Reviewer’s suggestion. We clarified the requested sentences in lines 594-598.
Point 54: Rephrase for clarification and also cite the underlying mechanisms briefly.
Response 54: Thank the Reviewer’s suggestion. We rephrased requested sentences and cited the underlying mechanisms as the Reviewer requested in lines592-598.
Point 55: I am sure that diabetes itself does not cause liver fibrosis. Please, rephrase, citing the underlying mechanisms and alterations related to diabetes that are associated with liver fibrosis occurrence in patients with NAFLD.
Response 55: Thank the Reviewer’s suggestion. We rephrased requested sentences and cited the underlying mechanisms as the Reviewer requested in lines 599-602.
Point 56: These sentences need clarification and better writing. Please, rephrase and improve the English quality.
Response 56: Thank the Reviewer’s suggestion. We rephrased and rewrote requested sentences as the Reviewer requested in lines 605-608.
Point 57: Exclude the ''the''. Rephrase giving examples of genes considered risk factors for NAFLD occurrence and progression.
Response 57: Thank the Reviewer’s suggestion. We rephrased, added examples of genes and cited appropriate publications as the Reviewer requested in lines 608-615.
Point 58: This is not scientifically written. Please, rephrase and explain it better.
Response 58: Thank the Reviewer’s suggestion. We rephrased and rewrote requested sentences as the Reviewer requested in lines 615-619.
Point 59: Avoid personal or possessive pronouns throughout the manuscript. Using these words is not scientific and decreases the academic rigor of the study.
Response 59: Thank the Reviewer’s suggestion. We rephrased and rewrote requested sentences as the Reviewer requested in lines 621-622.
Point 60: You cited right above that DM2 pathogenesis is complex. It would help if you did not mention it again here.
Response 60: Thank the Reviewer’s suggestion. We deleted the sentence as the Reviewer requested.
Point 61: Give examples and cite the underlying mechanisms.
Response 61: Thank the Reviewer’s suggestion. We added examples and cited appropriate publications as the Reviewer requested in lines 624-628.
Point 62: You can not generalize without a reference. Even with an appropriate reference, it is challenging to generalize. Please, reconsider.
Response 62: Thank the Reviewer’s suggestion. We rewrote the fragment to avoid generalized statement. The revised fragment is included in lines 628-635.
Point 63: References are missing in this whole paragraph. Additional references must be included for each sentence separately. Please, make the necessary corrections.
Response 63: Thank the Reviewer’s suggestion. The additional references were added as the Reviewer requested.
Point 64: Here, you cited vague examples without explaining the mechanisms that link inflammation to your subject matter. Please reconsider and rewrite.
Response 64: Thank the Reviewer’s suggestion. We rewrote the fragment as the Reviewer requested in lines 628-635.
Point 65: Delete.
Response 65: Thank the Reviewer’s suggestion. The sentence was deleted.
Point 66: Rewrite completely for clarification and better understanding.
Response 66: Thank the Reviewer’s suggestion. We rewrote the fragment as the Reviewer requested in lines 636-721.
Point 67: Please, avoid words like this.
Response 67: Thank the Reviewer’s suggestion. The word was changed to appropriate term as the Reviewer requested in lines 726-727.
Point 68: Rewrite completely for clarification and better understanding and also provide references.
Response 68: Thank the Reviewer’s suggestion. We rewrote the fragment and cited appropriate publications as the Reviewer requested in lines 727-733.
Point 69: Reconsider to ''many''.
Response 69: Thank the Reviewer’s suggestion. The word was changed to appropriate term as the Reviewer requested in line 735.
Point 70: Add an ''and''.
Response 70: Thank the Reviewer’s suggestion. The word was added as the Reviewer requested in line 736.
Point 71: Reconsider to ''depends''.
Response 71: Thank the Reviewer’s suggestion. The word was changed to appropriate term as the Reviewer requested in lines 737.
Point 72: Reconsider to ''5′-adenosine monophosphate (AMP)-activated protein kinase (AMPK)''.
Response 72: Thank the Reviewer’s suggestion. The term was changed as the Reviewer requested in line 737.
Point 73: Reconsider to ''increases''.
Response 73: Thank the Reviewer’s suggestion. The word was changed accordingly as the Reviewer requested in lines 741.
Point 74: I did not understand what you said here. Rewrite completely for clarification and better understanding.
Response 74: Thank the Reviewer’s suggestion. We rewrote the fragment as the Reviewer requested in lines 746-755.
Point 75: Specify ''liver'' comes to fibrosis.
Response 75: Thank the Reviewer’s suggestion. The term was changed as the Reviewer requested in line 751.
Point 76: What specific types of inflammatory processes? Your text studies the roles of metabolic disruptions leading to hepatic fibrosis. Please, be specific.
Response 76: Thank the Reviewer’s suggestion. We rewrote the fragment as the Reviewer requested in lines 746-752.
Point 77: Please, clarify better the hepatic fibrosis process and avoid misleading.
Response 77: Thank the Reviewer’s suggestion. We rewrote the fragment as the Reviewer requested in lines 746-752.
Point 78: Reconsider to ''It has been discovered that the induction of HSCs (Hepatic Stellate Cells) is associated with KCa3.1 and KCa2.3, which are components of a heterotetrameric voltage-independent potassium channel. Furthermore, it has been observed that oxidative stress upregulates these proteins, suggesting their involvement in the development of liver disease.''
Response 78: Thank the Reviewer’s suggestion. We rewrote the fragment as the Reviewer requested in lines 752-755.
Point 79: Add a ''the'' before ''association''.
Response 79: Thank the Reviewer’s suggestion. The word was added as the Reviewer requested in line 756.
Point 80: Add a comma here.
Response 80: Thank the Reviewer’s suggestion. A comma was added as the Reviewer requested in line 758.
Point 81: Reconsider to ''On the contrary, TGF-β1 plays a significant role in liver fibrosis by activating HSCs and promoting the generation of the extracellular matrix (ECM). However, it is essential to note that the activation of TGF-β1 can be counteracted or reduced by the presence of bone morphogenetic protein-7 (BMP-7). This suggests that BMP-7 may have a potential inhibitory effect on TGF-β1-mediated liver fibrosis.''
Response 81: Thank the Reviewer’s suggestion. We reconsidered the phrase as the Reviewer requested in lines 762-766.
Point 82: You used this expression twice in significantly closer lines. Please, reconsider this appearance.
Response 82: Thank the Reviewer’s suggestion. We reconsidered the phrase as the Reviewer requested in lines 762-766.
Point 83: Explain what this is and give the underlying mechanisms.
Response 83: Thank the Reviewer’s suggestion. We reconsidered the phrase as the Reviewer requested in lines 762-770.
Point 84: Rewrite for clarification.
Response 84: Thank the Reviewer’s suggestion. We rephrased this paragraph in line 766.
Point 85: Add a reference here.
Response 85: Thank the Reviewer’s suggestion. We added a reference in line 766.
Point 86: Reconsider to ''The investigation carried out by Duan et al. [62] did not establish definitive evidence regarding the connections between IFN-γ and NAFLD. However, the study did reveal that the administration of IFN-γ in a rodent model resulted in a reduction in liver fibrosis. This reduction exhibited a dose-dependent relationship and could potentially be linked to the suppression of ECM synthesis.''
Response 86: Thank the Reviewer’s suggestion. We reconsidered the phrase as the Reviewer requested in lines 766-770.
Point 87: Kindly ensure that this statement is accurately documented within the reference manuscript.
Response 87: Thank the Reviewer’s suggestion. We reconsidered the phrase as the Reviewer requested in lines 771-887.
Point 88: Reconsider to ''In comparison to healthy subjects, patients diagnosed with NAFLD were found to exhibit elevated levels of soluble vascular cell adhesion molecule-1 (sVCAM-1), which is closely associated with endothelial dysfunction.''
Response 88: Thank the Reviewer’s suggestion. We reconsidered the phrase as the Reviewer requested in lines 771-887.
Point 89: Avoid this expression.
Response 89: Thank the Reviewer’s suggestion. We deleted this expression in line 888.
Point 90: Kindly ensure that this statement is accurately documented within the reference manuscript.
Response 90: Thank the Reviewer’s suggestion. We added the correct reference manuscript in line 766-891.
Point 91: Please rephrase this statement to enhance its scientific value.
Response 91: Thank the Reviewer’s suggestion. We rephrased this paragraph in lines 892-893.
Point 92: Reconsider to ''Oxidative stress emerges as a pivotal factor of significance in liver function.''
Response 92: Thank the Reviewer’s suggestion. We reconsidered the phrase as the Reviewer requested in line 894.
Point 93: The content of this paragraph is irrelevant to the manuscript. Delete it.
Response 93: Thank the Reviewer’s suggestion. We deleted mentioned paragraph.
Point 94: Reconsider to only ''hepatic''.
Response 94: Thank the Reviewer’s suggestion. We reconsidered to “hepatic” as the Reviewer requested in line 904.
Point 95: Reconsider to ''Accumulating evidence suggests the involvement of hepatic B lymphocytes in the development of NAFLD and its progression towards NASH.''
Response 95: Thank the Reviewer’s suggestion. We reconsidered the phrase as the Reviewer requested in lines 904-905.
Point 96: Add a reference at the final of this sentence.
Response 96: Thank the Reviewer’s suggestion. We added a reference in line 907.
Point 97: Add a reference at the final of this sentence.
Response 97: Thank the Reviewer’s suggestion. We added a reference in line 908.
Point 98: Plural indicates ''cause''. Please, correct it.
Response 98: Thank the Reviewer’s suggestion. We corrected “causes” to the “cause” in line 910.
Point 99: Reconsider to ''Extensive molecular investigations have provided compelling evidence establishing a strong connection between gene expressions and the progression of NAFLD to NASH, ultimately leading to hepatic cirrhosis. Notably, specific genes such as IL-1β and TGFβ1, matrix metalloproteinase 9 (MMP9), matrix metalloproteinase 14 (MMP14), as well as ligands of chemokines including CCL2 and chemokine (C-X-C motif) ligand 1 (CXCL1), have been meticulously identified and associated with these pathological processes. Crucially, the regulation of these genes is governed by the NOD-like receptor protein 3 (NLRP3) inflammasome (NLRP3i). Activation of these genes triggers a cascade of events involving inflammatory processes, lipid metabolism regulation, and remodeling of the extracellular matrix.''
Response 99: Thank the Reviewer’s suggestion. We reconsidered the phrase as the Reviewer requested in lines 911-920.
Point 100: Add a reference at the final of this statement.
Response 100: Thank the Reviewer’s suggestion. We added a reference in line 922.
Point 101: Add a comma before the ''and''.
Response 101: Thank the Reviewer’s suggestion. We added a comma before “and” in line 923.
Point 102: Kindly rephrase the provided statement to augment its scientific merit.
Response 102: Thank the Reviewer’s suggestion. We rephrased this paragraph in lines 927-932.
Point 103: You can change to ''concerning''.
Response 103: Thank the Reviewer’s suggestion. We changed “in relation to” to “concerning” in line 932.
Point 104: Kindly rephrase the provided statement to augment its scientific merit.
Response 104: Thank the Reviewer’s suggestion. We rephrased this paragraph in lines 932-937.
Point 105: You can exclude this reference here. You mentioned Hasegawa et al. [74] right above.
Response 105: Thank the Reviewer’s suggestion. We excluded Hasegawa reference in line 937.
Point 106: You can consider ''The connection between C-C chemokine receptors type 2 (CCR2) and 5 (CCR5) and their ligands (CCL2 and CCL5) concerning liver fibrosis has been uncovered. These receptors and ligands are instrumental in triggering the inflammatory response and facilitating the migration of immune cells, ultimately leading to the development of hepatic fibrosis. However, significant advancements in treating liver fibrosis have been observed by administering an oral antagonist targeting CCR2 and CCR5 known as cenicriviroc. This treatment has demonstrated notable improvements in liver fibrosis and regression of systemic inflammatory markers, including IL-6, hs-CRP, IL-1β, and fibrinogen.'' Also, please include more references throughout the statement.
Response 106: Thank the Reviewer’s suggestion. We reconsidered the phrase as the Reviewer requested in lines 937-1109.
We sincerely hope that all changes introduced by us in the text will be fully satisfactory for the Reviewer.
Round 2
Reviewer 3 Report
The manuscript has been substantially improved. There are some typos and some missing punctuation marks. A final careful reading of the manuscript would be advisable, being generous with the commas.
Just a few examples :
- the sentence in line 288: Associated with NAFLD hepatic IR which may be... could be: Hepatic IR associated with NAFLD, which may be...
- exacerbateing in line 323 should be exacerbating
- there is a missing hyphen in domain containing 3 in line 335, domain-containing 3
- T2D in line 773, should be DM2, because this is the abbreviation used throughout the manuscript
Reviewer 4 Report
To the authors of manuscript ijms-2394791,
I sincerely thank you for incorporating the revisions in response to my initial comments on your manuscript. I have carefully reviewed all the amendments and considered the suggestions put forth by my fellow reviewer. Based on my evaluation, I am confident that your manuscript is now prepared for the final assessment by the MDPI team. My last comments are:
1) It still principally needs to be English-revised.
2)Please, also change the title to ‘’Type 2 Diabetes Mellitus, Non-Alcoholic Fatty Liver Disease and Metabolic Repercussions: The Vicious Cycle and the Interplay with Inflammation.’’ Change the last ‘’,’’ to a ‘’:’’.
Thank you once again for your diligent efforts in addressing the feedback and enhancing the quality of your work.
Best regards,
The Reviewer.
The manuscript still lacks a high-quality English review. I do not think the final English editing by MDPI during the final processing stages of the manuscript will be sufficient to assess all the quality enhancement the manuscript needs.